# Measuring coordination between women's self-help groups and local health systems in rural India: a social network analysis

Jenny Ruducha,[1] Divya Hariharan,[2] James Potter,[3] Danish Ahmad,[4,5] Sampath Kumar,[6] P S Mohanan,[6] Laili Irani,[7] Katelyn N G Long[8]

For numbered affiliations see end of article.

**Correspondence to**
Dr Jenny Ruducha;
jenny@braintreeglobalhealth.org

## ABSTRACT

**Objectives** To assess how the health coordination and emergency referral networks between women's self-help groups (SHGs) and local health systems have changed over the course of a 2-year learning phase of the Uttar Pradesh Community Mobilization Project, India.

**Design** A pretest, post-test programme evaluation using social network survey to analyse changes in network structure and connectivity between key individuals and groups.

**Setting** The study was conducted in 18 villages located in three districts in Uttar Pradesh, India.

**Intervention** To improve linkages and coordination between SHGs and government health providers by building capacity in leadership, management and community mobilisation skills of the SHG federation.

**Participants** A purposeful sampling that met inclusion criteria. 316 respondents at baseline and 280 respondents at endline, including SHG members, village-level and block-level government health workers, and other key members of the community (traditional birth attendants, drug sellers, unqualified rural medical providers, pradhans or elected village heads, and religious leaders).

**Main outcome measures** Social network analysis measured degree centrality, density and centralisation to assess changes in health services coordination networks at the village and block levels.

**Results** The health services coordination and emergency referral networks increased in density and the number of connections between respondents as measured by average degree centrality have increased, along with more diversity of interaction between groups. The network expanded relationships at the village and block levels, reflecting the rise of bridging social capital. The accredited social health activist, a village health worker, occupied the central position in the network, and her role expanded to sharing information and coordinating services with the SHG members.

**Conclusions** The creation of new partnerships between traditionally under-represented communities and local government can serve as vehicle for building social capital that can lead to a more accountable and accessible community health delivery system.

## Strengths and limitations of this study

► Original data to study and measure multisectoral co-ordination intervention between women's SHGs and local health systems.

► A detailed examination of health coordination and emergency referral networks in rural India across 18 villages and three districts.

► Contributes to literature as studies focusing on how economically marginalised women engage through SHGs in coordinating with government departments are not common.

► Limitations of social network analysis affects the ability to assess causality and generalisability.

► Limited duration of the linkage intervention during the Learning Phase may reduce ability to detect major changes, as a capacity building interventions take time.

## INTRODUCTION

### Background and study objective

Microfinance institutions comprising self-help groups (SHGs) are increasingly recognised as promising avenues for expanding health and social services to vulnerable populations.[1–3] In India, the concept of women's SHGs has evolved over the past three decades. The basic SHG structure remains defined as informal groups of 10–20 women from similar socio-economic backgrounds living in close proximity.[4] During the 1980s, the objectives were to engage women in collective savings activities and to provide access to credit. By the early 1990s, the official SHG and bank linkage programme in India were led by the National Agricultural Bank for Rural Development and focused on loans for livelihood activities. To improve scalability, by early 2000s, the SHG model grew into a key government programme providing financial access to the poor and addressing issues of social justice to improve the welfare of its members.[5]

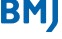

The evidence for supporting women's SHGs continues to grow as research demonstrates that increasing poor women's access to working capital can result in improvements in education and health,[2 6] and that SHGs tend to use their savings and credit for the family's well-being. Enhancing a woman's agency sets into motion new abilities to 'exercise bargaining power as well as develop a sense of self-worth, a belief in one's ability to secure desired changes, and the right to control one's life'.[7] As SHGs build social capital, they can also be instrumental in addressing deficits in government health systems.[8 9]

Building on the strengths of SHGs, a 5-year Uttar Pradesh Community Mobilization Project (UPCMP) was established in India in 2012 to implement a package of interventions to improve the practice of healthy maternal and child behaviours and to coordinate with the local government health system to improve maternal and newborn health. The project was based on activating an SHG model developed by Rajiv Gandhi Mahila Vikas Pariyojana (RGMVP), a non-governmental organisation, to expand SHGs into federated networks of village organisations and block organisations. Through capacity building and leadership training, poor and lower caste women were encouraged to access financial products, and to advocate for government health services and entitlements. The objective of our study was to assess how health services coordination and emergency referral networks between SHGs and local health systems, along with other key stakeholders, changed over the course of a 2-year learning phase of the project using social network analysis (SNA).

### Relevance of coordination networks

Coordination among community institutions is achieved through partnerships that improve responses to public and social issues[10] and are built around norms of reciprocity and trustworthiness.[11] The effectiveness of social networks is dependent on the density of community connections and the vibrancy of associations[12] to expand the relationships between diverse groups and to obtain the full range of knowledge, skills and resources that the community needs to solve complex problems. Bridging social ties is most effective between people and organisations from typically under-represented communities and groups with expertise that can provide access to schemes and services (such as frontline workers and doctors).[13]

### SHG's role in coordination networks

SHG platforms have reached 57.9% of villages in India, resulting in 4.8 million credit-linked groups in 2010, which demonstrate the broad potential for empowering communities to demand accountability from government functionaries.[1] Many social service models, such as the UPCMP, aim to promote coordination with government services to expand the exchange of resources and to generate social capital.[14–18] When community networks such as SHGs create linkages with government programmes and providers, more resources are available,

and together, these groups can tackle issues that no one group can resolve by itself.[19]

Participatory policies and community initiatives have been well studied and critiqued.[20–22] However, studies focusing on how economically marginalised women engage through SHGs in coordinating with government departments are difficult to find in the published literature. Our study aims to contribute to this research gap by examining the efforts of SHGs in coordinating the delivery of health services with the local government health system in order to build and sustain effective networks that enable the flow of resources to the poorest communities.

## METHODS

### Setting

Uttar Pradesh, at approximately 200 million population,[23] is one of the largest states in India constituting 16.5% of the country's total population. UPCMP began in 2012 with a learning phase in 100 administrative villages, called gram panchayats, located in 10 blocks within 8 districts with a goal to scale up the intervention to 120 blocks over the 5-year project period. During the learning phase of the project, we selected one block from three districts that represented different parts of the state (Hardoi, Mirzapur and Banda) (figure 1). In each block, we then selected 6 out of the 10 gram panchayats, for a total of 18 gram panchayats in our analysis.

### Linkage intervention

The linkage strategy aimed to improve coordination between SHGs and government health and social services by expanding the leadership, management and community mobilisation skills of the SHG federation at the village organisation and block organisation levels (box 1). The interventions focused on (1) facilitation of regular meetings of village organisation members with local government health workers and between the SHG block organisation with block-level health functionaries (box 1), (2) exchange of lists of pregnant and recently delivered women to promote early entry into prenatal and postnatal care; and (3) identification and dissemination of information about entitlements and emergency health facilities.

### Study design

We developed a pretest, post-test social network survey. The study objective was to assess how the health coordination and emergency referral networks between women's SHGS and local health systems changed over the course of the 2-year learning phase of the UPCMP. The survey instruments were based on a validated survey design structure,[24] and questions were developed and then pretested in a social and cultural setting similar to the study population to capture aspects of village-level and block-level connections that would be relevant for assessing the programme. One of the main survey questions asked all

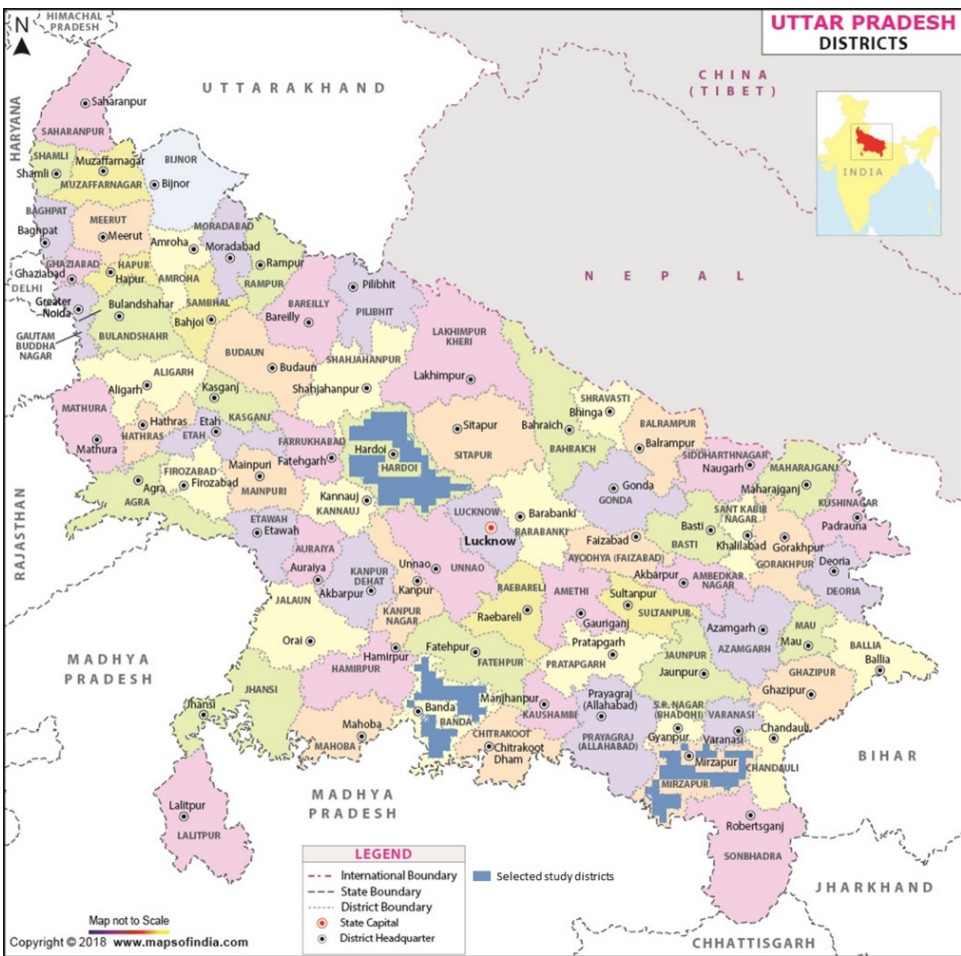

**Figure 1** Map of three study districts in Uttar Pradesh, India (source: https://mapsofindia.com; permission was granted by Compare Infobase, Ltd, New Delhi, India, to reproduce the map with adjusted blue shading to designate study districts).

respondents whether they coordinated health services, including emergency referrals, with every other respondent type in the survey (table 1).

### Study sample

The data were collected through a purposeful sampling methodology that included SHG members and those participating at the federated village organisation and block organisation levels, government and private health workers, RGMVP staff and other key stakeholders. Block-level respondents were interviewed only about their relationship with respondents from two of the six gram panchayats to reduce the length of the interviews for block-level respondents.

Certain roles are unique in a village or a block, such as the accredited social health activist (ASHA), the auxiliary nurse midwife (ANM), or the pradhan or village leader. For those roles, the inclusion criteria were that the respondent agrees to the interview, is over age 18 years, and is the person responsible for that role at its respective geographical level. For roles that had multiple potential representatives at a given level, the survey staff made a list of all potential respondents by consulting with staff at health facilities within each block, RGMVP programme staff and local stakeholders in each village, and attempted

to contact them in a random order. The first potential respondent who was successfully contacted was interviewed. The total sample was 596, with 316 respondents in the baseline and 280 in the endline for response rates of 94% and 82%, respectively.

Table 1 presents the complete list of respondents, along with their role designations, the acronyms used in SNA visual plot construction, village or block location, and a brief summary of their respective roles. The respondents were grouped into four broad categories corresponding to their affiliation: SHG structure, RGMVP staff, government health and nutrition and 'other' key stakeholders. This approach incorporates elements of both relational and positional models for examining networks of relations with the network structure.[25]

### Data collection

The baseline data were collected between November 2013 and January 2014, and endline surveys were adminstered between October and November 2015. Surveys were developed in CSPro V.6.0 and were adminstered using electronic tablets. During both data collection periods, the survey team spent 2 weeks in each of the three survey districts. Each team had a supervisor who monitored

## Box 1    Structure and roles of federated self-help groups

- ▶ *At community/neighborhood level*, 10–20 women from particularly vulnerable and marginalised households are organised into self-help groups (SHGs). These SHGs meet regularly for the purpose of addressing common problems through mutual support. In the case of Rajiv Gandhi Mahila Vikas Pariyojana, SHGs are responsible for promoting savings among groups, ensuring credit access from banks, and driving community-based social and behaviour change interventions associated with maternal, newborn and child health. As SHGs form in a village, two members from each SHG are voted into a village organisation whose members represent, on average, about 150–250 SHG women. Subsequently, two members from each village organisation across many villages are elected to the block organisation at the block level, representing 5000–7000 women.
- ▶ *At the village level*, village organisation members' main focus was to establish functional relationships between SHG federation members and the three local government health worker cadres designated as 'AAAs': the accredited social health activist (ASHA), a community health worker who gets paid based on her ability to mobilise pregnant and recently delivered women to seek recommended health services and to promote institutional deliveries; the auxiliary nurse midwife, a trained midwife who organises monthly health clinics in each village and supervises the ASHA; and the Anganwadi worker, a local nutrition worker who operates a day care center and distributes supplementary food for eligible children and pregnant and lactating women.
- ▶ *At the block level*, village organisation members are voted into the block organisation. Their roles were to develop relationships with the supervisors of the village-level health workers, medical staff working at the primary healthcare centres and other block-level government functionaries and elected officeholders.

data quality and a UPCMP project representative who provided logistical support.

### Analytical structure and network measures

In consideration of the models to guide network measurement and analysis at the individual and whole-network levels, our work fits into Mays and Scutchfield's[13] categorisation, including degree centrality, density and centralisation.[13] These measures are used to assess changes in health services coordination networks within the gram panchayat level and between key players comprising gram panchayat–block relationships. *Degree centrality* measures the number of connections or ties that each respondent maintains,[26] and our analysis is based on a mutual confirmation process. In other words, if one person acknowledges a relationship and the other person does not, that relationship is dropped from the network. Overall, 31% of ties remained after the confirmation process. The high loss of unconfirmed ties signals a weak level of connectivity, whereas ties confirmed by both parties have a higher probability of producing collaborative relationships.[26] In India, the caste system exerts barriers to relationship formation and contributes to a reduction in reciprocity of ties. *Density is* often used as a measure of social capital[11] and is defined as a ratio of existing relationships or ties in comparison to the potential number

of linkages.[27] *Centralisation* is an expression of how tightly the network structure is organised around its most central point. The general procedure is to calculate the differences between the centrality scores of the most central point and those of all other points to generate a ratio of the actual sum of differences to the maximum possible sum of differences.[27]

### Data analysis

SNA methods have been used to study the structural makeup of cooperation that can lead to stronger collaborative relationships.[28 29] The analysis used a combination of two software tools: R V.3 (www.r-project.org)[30] and UCINET V.6.[31] The plot visualisation was developed by using NetDraw.[32]

### Patient and public involvement

No patients or members of the public were involved in the development of research questions, the design of the study, or the development of outcome measures. Also, no patients were asked to advise on interpretation or writing up of results.

## RESULTS
### Descriptive

The characteristics of study respondents for the two study periods are presented in table 2. Most demographic indicators were balanced across baseline and endline rounds using a $\chi^2$ test, except for caste (p<0.001). For this indicator, scheduled caste and scheduled tribe accounted for a greater share of respondents in the baseline compared with the endline, while other backward caste representations were greater in the endline. The median age of the respondents was approximately 40 years. About a quarter to a third of all respondents were not educated. In both rounds of the survey, more than 50% of SHG-level respondents reported having no formal education, while all government health workers reported some education and approximately one-third reported postgraduate studies. Four district-level respondent roles were added in the endline.

Respondents were also asked whether they had friends or neighbours who were SHG members, and these results are presented in table 3. The percentage of SHGs in the federated structure that knew other SHG members between the baseline and endline increased significantly from 87% to 100%. The majority of RGMVP staff respondents knew SHG members in their gram panchayat or village. Only about one-third of government health workers knew an SHG member, while almost two-thirds of respondents from the 'other' category (including the pradhan (local leader), drug shop owner and traditional birth attendant (TBA)) knew an SHG member.

### Social network analysis

A fixed set of gram panchayat and block-level providers and other key community members involved in

**Table 1** Study respondents defined

| Acronym (used in SNA plots) | Respondent | Definition |
|---|---|---|
| **SHG structure** | | |
| GM | SHG member | Part of SHG group, involved in saving and borrowing, weekly SHG meetings. |
| G1SS and G2SS | SHG swasthya sakhi (two per village) | SHG member who has received special training in health. |
| VOSS | Village organisation's swasthya sakhi | SHG leader at village level trained in health and linkages strategy to coordinate health activities. |
| VOM and VOM2 | Village organisation member (two per village) | Leader from two village VOs trained in linkages strategy to coordinate health activities. |
| VOB | Village organisation office bearer | SHG member who holds a post at the federation of SHGs at the village level called VO. |
| MS | Meeting sakhi | SHG member who organises weekly SHG meetings. |
| BOB | Block organisation office bearer | SHG and VO member who holds a post at the federation level called BO. |
| BOR | Block organisation representative from VO | A member of the BO representing her village organisation. |
| BOPR | Block organisation poverty reduction committee member | A dedicated committee in the BO responsible for development strategies on mobilising women from poor households into SHGs. |
| **RGMVP staff** | | |
| RG | RGMVP | Full time field staff who works for the non-government organisation (RGMVP) at block level. |
| RGF | RGMVP field officer | Full-time field staff who acts as a link between RGMVP and SHGs. |
| RGV | RGMVP volunteer (community volunteer) | A part-time volunteer from the community who supports the full-time field staff in implementation activities. |
| RGT | RGMVP trainer | RGMVP professional trainer who conducts field trainings for SHGs. |
| ISC | Internal social capital | Nominated SHG member who is trained to support health implementation in villages by facilitating discussions in various SHG meetings. |
| **Government health and nutrition** | | |
| ASHA | Accredited social health activist | Community health volunteer paid honorarium to promote basic preventative maternal and child services and is supervised by ANM. |
| ANM | Auxiliary nurse midwife | Government-trained community health worker who conducts monthly outreach clinics providing maternal and child care. ANM supervises ASHA and works with AWW. |
| AWW | Anganwadi worker | Government community worker providing food supplements to young children, adolescent girls and lactating women, along with preschool child education services in government-operated village creches. |
| ANMS/LHV | Auxiliary nurse midwife supervisor/female health visitor | ANMs who have been promoted to oversee six subcentres covering a population of 30 000. |
| ICDSs | Integrated child development scheme supervisor | Government worker appointed who supervises 25 village-level creches called Anganwadi Centres where the AWW works. |
| CDPO | Community development programme officer | Government worker in charge of the ICDS Project at block level who oversees the ICDS supervisor at block level. |
| BDO | Block development officer | Government social worker appointed under the ICDS scheme and entrusted with the overall responsibility of the ICDS at block level. |
| PHM | Primary health centre medical officer | Government medical doctor and primary administrator of primary health centres. |
| PHN | Primary health centre nurse | Government nurse providing nursing services in primary health centres. |
| PHC | Primary health centre staff | Approximately 13 government staff appointed to administer primary care services at primary health centres. |
| PHS | PHC other | Extra support staff appointed at high-delivery load primary health centres. |
| CHN | Community health centre nurse | Government nurse appointed at community health centres, which constitute the secondary level of healthcare. |
| CHM | Community health centre medical officer | Government medical officer who works at the community health centre to provide clinical services. |
| CHC | Community health centre staff | The total strength of staff (approx. 46) appointed by the government to administer specialist and referral care services at the community health centre. |

| Table 1 | Continued | |
|---|---|---|
| **Acronym (used in SNA plots)** | **Respondent** | **Definition** |
| CMO | Chief medical officer-in-charge | Government medical doctor who works as the programme director at the district level. |
| CHS | Community health centre other | Extra staff appointed at high patient load community health centres. |
| DHN | District hospital nurse | Government nurse working in a 300-bed district hospital for providing comprehensive secondary healthcare. |
| DHM | District hospital medical officer | Government medical officer who works at the district hospital to provide clinical services. |
| DHOB | District hospital OB/GYN | Government medical specialist appointed at the district hospital to provide specialist OB/GYN care for women. |
| RKS | Rogi Kalyan Samiti | A patient welfare committee set-up at the government hospital to ensure accountability and protection of patients' rights. |
| **Other stakeholders** | | |
| PRI | Pradhan or village head | Elected head of a village-level constitutional body of local self-government called the panchayat, who acts as focal point of contact between government officers and the village community. |
| RL | Religious leader | Leader of a religion recognised at the community (village) level. |
| RMP | Rural medical practitioner (unqualified) | An unqualified medical practitioner who is not formally trained in providing healthcare services in villages. RMPs are culturally recognised and are often the first point of healthcare in communities. |
| Dr | Medical doctor (qualified) | Formally trained and accredited medical doctor that provides healthcare. |
| PF | Private health facility provider | Accredited medical doctor providing health services in a private health facility. |
| DS | Drug shop owner | Owner of a private chemist shop in villages who dispenses drugs and health advice |
| BPRI | Block Panchayat Raj Institution | The middle tier of the system of local self-government in India operating at the administrative level of block that links villages with districts. |

*TSU project workers.
BO, block organisation; ICDS, integrated child development scheme; OB/GYN, obstetrics and gynaecology; RGMVP, Rajiv Gandhi Mahila Vikas Pariyojana; SHG, self-help group; SNA, social network analysis; TSU, technical support unit; TSUB, TSU block coordinator; TSUC, TSU community specialist; TSUN, TSU nurse mentor; VO, village organisation.

healthcare delivery was included in the analysis. Network measures for density, centralisation and average degree centrality are presented individually for all 18 villages (6 per district), as well as the average scores across each district for the baseline and endline (table 4). However, since block-level respondents were asked about relationships with only two gram panchayats, we confirmed whole gram panchayat–block relationships for these two gram panchayat–block dyads in each of the three blocks, along with results for baseline and endline (table 5).

### Network metrics

The density of the overall village health services and referral networks in 18 gram panchayats (villages) were low but increased from baseline to endline in two out of three districts: Banda (7.5%–10.1%) and Mirzapur (4.7%–12.6%). Hardoi had a higher density at baseline compared with the other two blocks and had a slight reduction at endline (13.6%–11.4%). While the average patterns of density change in blocks during the 2-year intervention period increased, variability was noted beween individual villages in each of the blocks related to their baseline characteristics, and the degree of change that occurred between baseline and endline. Although Mirzapur's density was the lowest in the gram panchayats

measured, they had the greatest growth, as noted by tripling of their scores during the 2-year UPCMP intervention. Hardoi had a reduction in density to half of gram panchayats but remained within a close range to the other two blocks at endline.

The gram panchayat–block level health services and referral networks took into account the relationships between two villages and block level respondents within each block (table 5). The baseline densities were lower overall than village-level networks but increased after 2 years of project implementation, although the level of increase was less than in the village-level networks. Similar to village-level network patterns, Hardoi's gram panchayat–block level health services and emergency referral network remained at the highest level in comparison with the other two blocks in both baseline and endline, but decreased slightly in the endline (12.4–10.5).

The centralisation measures increased two- to threefold during the same 2-year intervention period, with the highest level of increase in Mirzapur gram panchayats, which doubled on average (15.1%–29.2%). Within the overall average increase in all blocks, a gram panchayat-level variation was noted. This included a decrease in centralisation in two Banda villages and four Hardoi

**Table 2** Comparison of respondent background information

| Characteristic | UPCMP baseline, 2013 (Banda, Mirzapur and Hardoi) n (%) | UPCMP endline, 2015 (Banda, Mirzapur and Hardoi) n (%) |
|---|---|---|
| Median age | 41 | 40 |
| Education | | |
| No education | 100 (32) | 74 (26) |
| Some primary (up to 8th standard) | 63 (20) | 74 (26) |
| Some secondary (up to 11th standard) | 40 (13) | 40 (14) |
| Secondary completed (12th standard) | 2 (1) | 3 (1) |
| Postgraduate (at least some college completed) | 111 (35) | 89 (32) |
| Caste 1* | | |
| Scheduled caste/scheduled tribe | 172 (55) | 106 (38) |
| Other backward castes | 71 (22) | 112 (40) |
| General caste | 71 (22) | 62 (22) |
| Affiliation | | |
| SHG structure | 119 (38) | 98 (35) |
| RGMVP staff | 23 (7) | 24 (9) |
| Government health | 77 (24) | 77 (28) |
| Other | 97 (30) | 81 (29) |
| Location | | |
| Purwa (hamlet) | 34 (11) | 33 (12) |
| Gram panchayat (village) | 220 (70) | 203 (73) |
| Block | 62 (20) | 34 (12) |
| District | 0 | 10 (4) |
| Total | 316 | 280 |

The classification of castes is formalised by the Government of India into these categories, and we used standard definitions to create these categories. Some of these lower caste designations enable caste groups to receive specific government benefits and subsidies. Respondents were asked to self-identify into caste categories in the survey.

*$\chi^2$ test: p<0.001.

RGMVP, Rajiv Gandhi Mahila Vikas Pariyojana; SHG, self-help group; UPCMP, Uttar Pradesh Community Mobilization Project.

villages. Centralisation in the gram panchayat–block networks also increased in all three blocks with the largest gains in Mirzapur (8.9%–23.9%), while Banda increased from 18.5% to 29.0% and Hardoi from 28.9% to 30.1%.

Mirzapur and Banda had the highest increase in average degree centrality, but endline results suggested similar levels of provider, SHG, key health provider and community contacts in all three blocks with 3.1–3.6 contacts per

**Table 3** Respondent with friends or neighbours who are SHG members by 'affiliation' (row percentages)

| Affiliation | UPCMP, 2013 (Banda, Mirzapur and Hardoi) n=316 | | | UPCMP, 2015 (Banda, Mirzapur and Hardoi) n=280 | | |
|---|---|---|---|---|---|---|
| | No (%) | Yes (%) | Don't know (%) | No (%) | Yes (%) | Don't know (%) |
| SHG structure* | 16 (13) | 103 (87) | 0 (0) | 0 (0) | 98 (100) | 0 (0) |
| RGMVP staff | 4 (17) | 18 (78) | 1 (4) | 1 (4) | 23 (96) | 0 (0) |
| Government health | 45 (58) | 23 (30) | 9 (12) | 46 (60) | 26 (34) | 5 (6) |
| Other | 33 (34) | 58 (60) | 6 (6) | 22 (27) | 53 (65) | 6 (7) |
| Total | 95 (31) | 202 (64) | 16 (5) | 69 (25) | 200 (71) | 11 (4) |

*Fischer's exact test: p value<0.0001. All other rows are insignificant.

SHG, self-help group; UPCMP, Uttar Pradesh Community Mobilization Project.

**Table 4** Sociometric gram panchayat level coordination and referral networks: confirmed density, centralisation and average degree centrality

| District (one block per district) | Confirmed density (%) | | Confirmed centralisation (%) | | Average degree centrality | |
|---|---|---|---|---|---|---|
| | Baseline | Endline | Baseline | Endline | Baseline | Endline |
| Banda gram panchayats | | | | | | |
| 1 | 10.0 | 7.5 | 17.8 | 13.3 | 3.0 | 2.3 |
| 2 | 5.8 | 8.3 | 15.1 | 33.8 | 1.8 | 2.5 |
| 3 | 10.8 | 16.7 | 31.1 | 39.1 | 3.3 | 5.0 |
| 4 | 4.2 | 5.8 | 16.9 | 15.1 | 1.3 | 1.8 |
| 5 | 7.5 | 13.3 | 20.4 | 28.4 | 2.3 | 4.0 |
| 6 | 6.7 | 9.2 | 14.2 | 18.7 | 2.0 | 2.8 |
| Average | 7.5 | 10.1 | 19.3 | 24.7 | 2.3 | 3.1 |
| Hardoi gram panchayats | | | | | | |
| 7 | 11.7 | 4.4 | 23.1 | 8.6 | 3.5 | 1.4 |
| 8 | 19.2 | 19.1 | 36.4 | 32.8 | 5.8 | 6.1 |
| 9 | 21.7 | 15.4 | 26.7 | 23.4 | 6.5 | 4.9 |
| 10 | 6.7 | 8.1 | 14.2 | 37.9 | 2.0 | 2.6 |
| 11 | 11.7 | 6.6 | 16.0 | 19.5 | 3.5 | 2.1 |
| 12 | 10.8 | 14.7 | 31.1 | 30.9 | 3.3 | 4.7 |
| Average | 13.6 | 11.4 | 24.6 | 25.5 | 4.1 | 3.6 |
| Mirzapur gram panchayats | | | | | | |
| 13 | 5.8 | 11.7 | 15.1 | 30.2 | 1.8 | 3.5 |
| 14 | 2.5 | 8.3 | 11.6 | 33.8 | 0.8 | 2.5 |
| 15 | 5.8 | 12.5 | 15.1 | 22.2 | 1.8 | 3.8 |
| 16 | 3.3 | 19.2 | 10.7 | 50.7 | 1.0 | 5.8 |
| 17 | 8.3 | 8.3 | 26.7 | 19.6 | 2.5 | 2.5 |
| 18 | 2.5 | 15.8 | 11.6 | 18.7 | 0.8 | 4.8 |
| Average | 4.7 | 12.6 | 15.1 | 29.2 | 1.5 | 3.8 |

**Table 5** Sociometric GP–block level coordination and referral networks: confirmed density, centralisation and average degree centrality

| District (one block per district) | Confirmed density (%) | | Confirmed centralisation (%) | | Average degree centrality | |
|---|---|---|---|---|---|---|
| | Baseline | Endline | Baseline | Endline | Baseline | Endline |
| Banda | | | | | | |
| GP 2+block | 7.1 | 7.4 | 12.7 | 34.3 | 4.4 | 4.7 |
| GP 5+block | 5.4 | 8.3 | 24.3 | 23.6 | 3.4 | 5.3 |
| Average | 6.3 | 7.9 | 18.5 | 29.0 | 3.9 | 5.0 |
| Hardoi | | | | | | |
| GP 8+block | 11.1 | 10.9 | 25.2 | 34.2 | 6.9 | 7.4 |
| GP 9+block | 13.7 | 10.1 | 32.5 | 26.0 | 8.5 | 6.9 |
| Average | 12.4 | 10.5 | 28.9 | 30.1 | 7.7 | 7.2 |
| Mirzapur | | | | | | |
| GP 13+block | 3.4 | 8.1 | 9.8 | 23.8 | 2.1 | 5.2 |
| GP 14+block | 2.0 | 8.0 | 7.9 | 24.0 | 1.3 | 5.1 |
| Average | 2.7 | 8.1 | 8.9 | 23.9 | 1.7 | 5.2 |

GP, gram panchayat.

person. Mirzapur started at the lowest level, with only 1.5 confirmed contacts, compared with Banda, with 2.3, and Hardoi, with 4.1, with variability at the individual village level ranging from 0.8 to 6.5 across the three blocks. All gram panchayat–block level measures had higher levels of contacts at baseline compared with gram panchayat levels with Hardoi reporting a high of 7.7 contacts. Endline average degree centrality measures in gram panchayat–block dyads in Banda, Mirzapur and Hardoi were 5.0, 5.2 and 7.2, respectively.

## Health services and emergency referral networks

The structures of the health services coordination and emergency referral networks were visualised for all 18 gram panchayats (villages) and for the 6 gram panchayat–block dyads. We selected one example from each gram panchayat–block network to illustrate the diversity of changes that occurred in structure and network connectivity. The nodes were labelled by respondent, color coded to reflect the type of respondent group and shaped according to the location of the respondent's usual residence or work location. The nodes were sized by betweeness centrality, with larger nodes signifying the increasing importance of that respondent in linking others that may not be directly connected to each other.

The structure and the relationships between different individuals and groups changed between baseline and after 2 years of UPCMP implementation. At baseline,

the Mirzapur gram panchayat–block health services and referral network had a 'kite-like' structure, with only two of the four respondent groups in the network (government health and social workers and the SHG structure) having relationships with no block-level connections (figure 2A). At one end, a triad of the three village health and social workers (ASHA, ANM and Anganwadi worker (AWW)) were connected together. The AWW had a relationship with an SHG member but had no direct connections with the SHG federated structure. The SHG structures themselves (at the tail end of the kite-like structure) were not integrated. At endline (figure 2B), the network structure expanded into a series of connected sets of relationships with the expansion of the number of participants and the inclusion of the TBA and RGMVP's block-level internal social capital worker. The shift from the AWW, a role more focused on nutrition and child development, to the ASHA as the main connector to the SHG structure at two critical points, directly to an SHG member, as well as the SHG village organisation's swasthya sakhi (VOSS), expanded the potential for better health services coordination. The SHG structure also became better integrated with the village organisation-level functionaries linked together with the exception of the VOSS.

At baseline, the Banda gram panchayat–block level health services and referral network was organised into two small worlds, with the SHG structure and the RGMVP

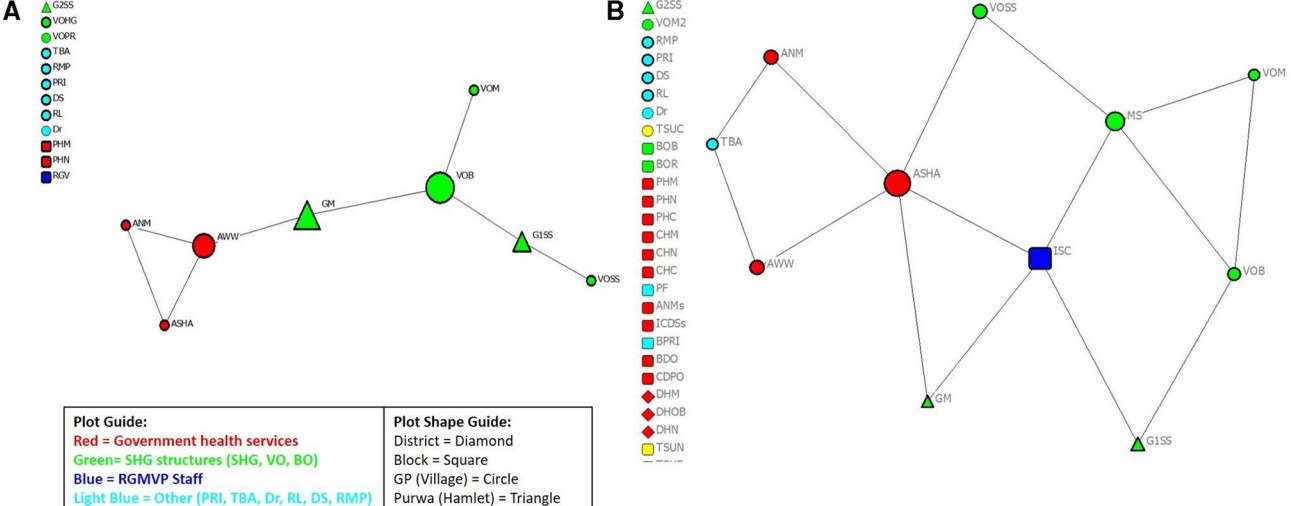

**Figure 2** (A) Mirzapur GP–block coordination network: baseline. (B) Mirzapur GP–block coordination network: endline. ANM, auxiliary nurse midwife; ASHA, accredited social health activist; AWW, Anganwadi worker; BDO, block development officer; BO, block organisation; BOB, block organisation office bearer; BOR, block organisation representative from VO; BPRI, block Panchayat Raj Institution; CDPO, community development programme officer; CHC, community health centre staff; CHM, community health centre medical officer; CHN, community health centre nurse; DHM, district hospital medical officer; DHN, district hospital nurse; DHOB, district hospital OB/GYN; Dr, medical doctor (qualified); DS, drug shop owner; G1SS/G2SS, SHG swasthya sakhi; GM, SHG member; GP, gram panchayat; ICDSs, integrated child development scheme supervisor; ISC, internal social capital; MS, meeting sakhi; PF, private health facility provider; PHC, primary health centre staff; PHM, primary health centre medical officer; PHN, primary health centre nurse; PRI, pradhan or village head; RGMVP, Rajiv Gandhi Mahila Vikas Pariyojana; RGV, RGMVP volunteer (community volunteer); RL, religious leader; RMP, rural medical practitioner; SHG, self-help group; TBA, traditional birth attendant; TSUC, TSU community specialist; TSUN, TSU nurse mentor; VO, village organisation; VOB, village organisation office bearer; VOM, village organisation member; VOSS, village organisation's swasthya sakhi; VOPR, village organisation poverty reduction committee member; VOHG, village organisation health and gender committee member.

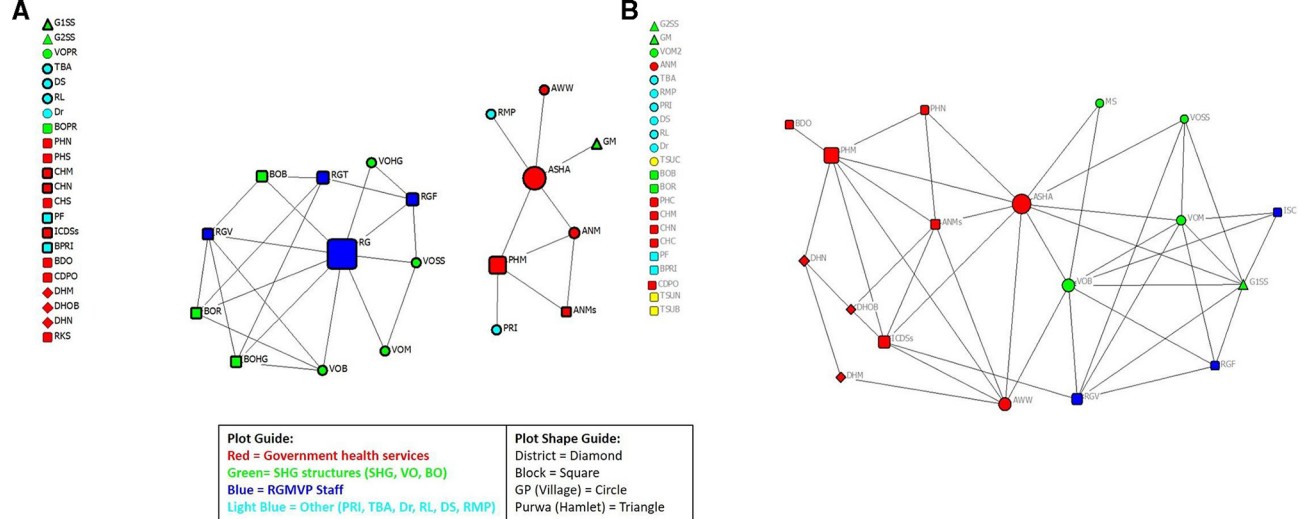

**Figure 3** (A) Banda GP–block coordination network: baseline. (B) Banda GP–block coordination network: endline. ANM, auxiliary nurse midwife; ANMs, auxiliary nurse midwife supervisor; ASHA, accredited social health activist; AWW, Anganwadi worker; BDO, block development officer; BO, block organisation; BOB, block organisation office bearer; BOHG, block organisation health and gender committee member; BOPR, block organisation poverty reduction committee member; BOR, block organisation representative from VO; BPRI, block Panchayat Raj Institution; CDPO, community development programme officer; CHC, community health centre staff; CHM, community health centre medical officer; CHN, community health centre nurse; DHM, district hospital medical officer; DHN, district hospital nurse; DHOB, district hospital OB/GYN; Dr, medical doctor (qualified); DS, drug shop owner; G1SS/G2SS, SHG swasthya sakhi; GM, SHG member; GP, gram panchayat; ICDSs, integrated child development scheme supervisor; ISC, internal social capital; MS, meeting sakhi; PF, private health facility provider; PHC, primary health centre staff; PHM, primary health centre medical officer; PHN, Primary primary Healthhealth centre nurse; PRI, pradhan or village head; RGMVP, Rajiv Gandhi Mahila Vikas Pariyojana; RGV, RGMVP volunteer (community volunteer); RKS, Rogi Kalyan Samiti; RL, religious leader; RMP, rural medical practitioner; SHG, self-help group; TBA, traditional birth attendant; TSUC, TSU community specialist; TSUN, TSU nurse mentor; VO, village organisation; VOB, village organisation office bearer; VOHG, village organisation health and gender committee member; VOPR, village organisation poverty reduction committee member; VOM, village organisation member; VOSS, village organisation's swasthya sakhi.

SHG support structure in one cluster, while the government health system and a few other key village respondents were in another separate cluster (figure 3A). Within the SHG small world cluster, RGMVP was at the centre of a centralised system resembling a 'spoke and wheel' with members of the SHG structure at the village and block levels connecting to RGMVP but not to each other.

In the second cluster, the ASHA was a leading connector of village-level respondents (AWW and the rural medical practitioner, an unqualified provider and an SHG member) to the broader government health system at the block level (ANM supervisor, primary health centre medical officer and the village-oriented ANM provider). By the endline, the two separate clusters formed one network (figure 3B). The ASHA became the main connector between the government health providers and the SHG structure. The SHG village organisation members also increased coordination within their own network and RGMVP. RGMVP scaled back from their centralised role at baseline and moved to the periphery of the network, while SHG enagagment increased.

In Hardoi, the structure remained similar since they started with one whole network; however, an additional project, the Technical Support Unit, a supply side intervention working with the health system, was just starting during the endline collection period (figure 4A,B). The

ANM held the central position in the endline health services and emergency referral coordination network, and acted as an intermediary between the government health functionaries at the block level and the SHG federated structure. The ASHA also stood out as more direct connectivity was established with different members of the SHGs. Within the SHG platform itself, there were more channels of direct communication between different members and women in designated leadership positions.

## DISCUSSION

After 2 years of UPCMP project implementation, the health services coordination and emergency referral networks increased in their density, but absolute levels remained low. This was a positive direction as higher density is considered an overall indicator of cohesion and interaction within a network and is often associated with faster rates of information diffusion within a community.[11] As information is more quickly shared, processed and appropriated, it can lead to shared decision making.[33]

However, our study results have limitations related to the constraints of SNA, minimal prior research on coordination networks between SHGs and government, and limited duration of testing the full effects of the linkage intervention. SNA has been mainly used for descriptive

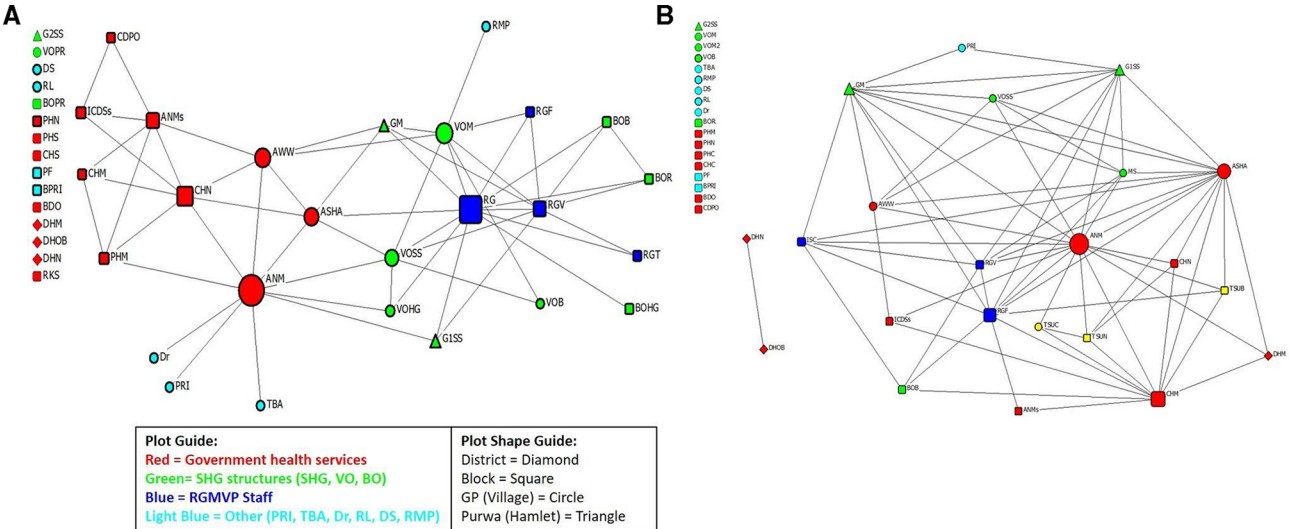

**Figure 4** (A) Hardoi GP–block coordination network: baseline. (B) Hardoi GP–block coordination network: endline. ANM, auxiliary nurse midwife; ANMs, auxiliary nurse midwife supervisor; ASHA, accredited social health activist; AWW, Anganwadi worker; BDO, block development officer; BO, block organisation; BOB, block organisation office bearer; BOHG, block organisation health and gender committee member; BOPR, block organisation poverty reduction committee member; BOR, block organisation representative from VO; BPRI, block Panchayat Raj Institution; CDPO, community development programme officer; CHC, community health centre staff; CHM, community health centre medical officer; CHN, community health centre nurse; DHM, district hospital medical officer; DHN, district hospital nurse; DHOB, district hospital OB/GYN; Dr, medical doctor (qualified); DS, drug shop owner; G1SS/G2SS, SHG swasthya sakhi; GM, SHG member; GP, gram panchayat; ICDSs, integrated child development scheme supervisor; ISC, internal social capital; MS, meeting sakhi; PF, private health facility provider; PHC, primary health centre staff; PHM, primary health centre medical officer; PHN, Primary primary Healthhealth centre nurse; PRI, pradhan or village head; RGMVP, Rajiv Gandhi Mahila Vikas Pariyojana; RGV, RGMVP volunteer (community volunteer); RKS, Rogi Kalyan Samiti; RL, religious leader; RMP, rural medical practitioner; SHG, self-help group; TBA, traditional birth attendant; TSUC, TSU community specialist; TSUN, TSU nurse mentor; VO, village organisation; VOB, village organisation office bearer; VOHG, village organisation health and gender committee member; VOM, village organisation member; VOPR, village organisation poverty reduction committee member; VOSS, village organisation's swasthya sakhi.

purposes and has been less frequently used for evaluating interventions,[34] and therefore needs further testing to determine broader generalisability. Sources of bias may include using purposive sampling to identify the respondent roles in the villages and blocks. Additionally, there are no established criteria for evaluating networks[35] as SNA represents a relatively new multidisciplinary methodology with limited empirical studies. While there is evidence in the literature that the constructs of density and network centrality may influence partnership functioning and coordination, their magnitudes and mechanisms of effect in public health are still largely unknown.[13] Our study provides more empirical evidence to further explore the potential of SNA to measure coordination networks.

Our study demonstrated not only an increase in connections between individuals but also an expansion of relationships between groups. Poor and traditionally lower caste SHGs developed skills to cross boundaries and to forge relationships with health providers. Traditionally bound by societal and cultural expectations, it has been difficult to overcome social and structural barriers that limit interactions among heterogenous groups. A strategy based on increasing diversity in relationships while actively working to reduce redundancy can lead to improved levels of bridging social capital in a network.

Such an approach that connects essential groups can lead to better ways of coordinating and collaborating.[11] Therefore, SHGs with increasing skills and confidence in engaging with health services providers are creating inroads to better health services for themselves and their communities.

As proxies for bridging social capital, the average degree centrality increased by creating new connections within the 2 years of UPCMP implementation. However, simply increasing the number of ties may not necessarily result in strong relationships, as noted by the weak ties theory.[36] Although it is common to surround ourselves with strong ties that include people similar to us in beliefs, values and access to resources, it is through weak ties that we begin to diversify our networks and create avenues for accessing more varied resources.[11] It is probable that building SHG leadership and coordination skills, which was a major UPCMP intervention, facilitated a pathway for SHGs to create weak ties with the health system. These patterns reflect opportunities for diverse stakeholders to engage in the planning and production of change.[37]

Our analysis showed that the ASHA occupied the central position in the network, and her role expanded to coordinating services with the VOSS and others in the SHG federation. The ASHA is a community health volunteer, and her role evolved from an 'activist' and

advocate of her village to being part of the government health system and being accountable to the ANM or nurse midwife, as well as the pradhan (leader) in the village health, nutrition and sanitation committee. As an important and recognised member of the community, the ASHA's modelling of cross-caste and class relationships can influence the governance dynamics as building collaborative networks are considered a more democratic means of developing public policy.[38–41]

## CONCLUSIONS

The SHG platform and its federated structure were developed by RGMVP to scale up and galvanise SHG development to reduce poverty, support women's empowerment and break caste-based hierarchies to encourage comprehensive rural development. The endline SNA reveals an expansion of pathways to coordinate health services and emergency referrals for poor, illiterate and low-caste women through the added voices of SHG members. The ability of women with multigenerational social and economic deprivations to broaden their exposure to social networks through SHG membership is a major step towards gaining self-confidence in participatory community development.

The increasing success of SHG federations to forge linkages with the health system leads to greater coordination for health services delivery while stimulating a more centralised structure of core village health workers. Collaborative processes that include individual empowerment, bridging social ties and synergy can strengthen the capacity to solve problems.[42] These ideas feed into the concept of state–society synergy, where a mobilised civil society and an active government can work together to build social capital and enhance each other's development efforts.[11 43]

If successful, these partnerships can serve as vehicles for transforming public health from a diverse collection of activities and organisations into an organised and accountable delivery system.[13] The challenge will be in maintaining these networks, so that they remain dynamic and offer new benefits to partners.[24] Future studies that include longitudinal data can provide deeper understanding of the mechanisms by which intersectoral partnerships and community mobilisation lead to effective coordination networks to tackle health problems and their socioeconomic determinants.

### Author affiliations
[1]Braintree Global Health, Cambridge, Massachusetts, USA
[2]Institute for Financial Management and Research, Chennai, Tamil Nadu, India
[3]Department of Global Health and Population, Harvard T.H. Chan School of Public Health, Boston, Massachusetts, USA
[4]Centre for Research and Action in Public Health, University of Canberra, Canberra, Australian Capital Territory, Australia
[5]Indian Institute of Public Health, Public Health Foundation of India, Gandhinagar, Gujarat, India
[6]Rajiv Gandhi Mahila Vikas Pariyojana, Raebareli, Uttar Pradesh, India
[7]Population Council, New Delhi, India
[8]Department of Global Health, Boston University School of Public Health, Boston, Massachusetts, USA

**Acknowledgements** The authors thank Dileep Mavalankar for his leadership of the Uttar Pradesh Community Mobilization Project; Deborah Maine, formerly at Boston University Center for Global Health and Development (BU CGHD), who supported the study's new methods; and ME Khan, formerly of the Population Council, for his knowledge and open discussions, which led to many ideas about how to study rural health systems. Many staff members of the Rajiv Gandhi Mahila Vikas Pariyojana, including Gyanendra Verma and Kapil Patil, and the Public Health Foundation of India (Anuraag Chaturvedi, Sanjit Sarkar and Srinivasan Soundararajan) assisted in the process of implementing this research project. We also acknowledge the administrative support and data collection oversight provided by Ariel Falconer, formerly of BU CGHD, for the baseline and Steven Crimaldi of BU CGHD for the endline. Lastly, we would like to thank the study respondents who offered their time and information to make this research possible.

**Contributors** Conceptualisation and project design: JR. Methodology: JR and JP. Network analysis: JP. Qualitative analysis: JP, JR and DH. Interpretation: JR, JP and DH. Interviewer training: JP, DH and JR. Field investigation: DH, DA, SK, PSM and JP. Draft preparation: JR. Literature review: JR. Revising and editing: JR, DH, LI, JP, KNGL and DA.

**Funding** This work was supported by the Bill and Melinda Gates Foundation (grant number OPP 1033910).

**Map disclaimer** The depiction of boundaries on the map(s) in this article do not imply the expression of any opinion whatsoever on the part of BMJ (or any member of its group) concerning the legal status of any country, territory, jurisdiction or area or of its authorities. The map(s) are provided without any warranty of any kind, either express or implied.

**Competing interests** None declared.

**Patient and public involvement statement** No patients or members of the public were involved in the development of research questions, the design of the study or the development of outcome measures. No patients were asked to advise on interpretation or writing up of the results.

**Patient consent for publication** Not required.

**Ethics approval** The surveys were approved by the Institutional Review Board (IRB) of Boston University Medical Campus (IRB number H-32603). A consent form (approved by Boston University Ethics Review Board) was read to participants and required a verbal approval.

**Provenance and peer review** Not commissioned; externally peer reviewed.

**Data availability statement** Data are available upon reasonable request.

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
