## [Reviewer comments · BMJ Open]

ARTICLE DETAILS

TITLE (PROVISIONAL)	Measuring Coordination between Women's Self-Help Groups and Local Health Systems in Rural India: A Social Network Analysis
AUTHORS	Ruducha, Jenny; Hariharan, Divya; Potter, James; Ahmad, Danish; Kumar, Sampath; Mohanan, P.S.; Irani, Laili; Long, Katelyn

VERSION 1 - REVIEW

REVIEWER	Candace Forbes Bright East Tennessee State University, USA
REVIEW RETURNED	15-Feb-2019

GENERAL COMMENTS	This is a fascinating study and paper and I have enjoyed the opportunity to review it. It is well-developed, but I think there are ways to improve it: 1) A broader introduction is needed. Before the getting into the role of SHGs, it would be helpful to understand their context. More information on what they are needed, how exactly they function, and any comparable initiatives would be helpful from the beginning to better understand and assess the rest of the study. Most importantly, I don't feel like I ever quite understood who was participating in the SHG and how they functioned. The methods did not clarify this either. The information in Box 1 is helpful for understanding the relationship between SHGs and VOs, but a similar introduction is needed for SHGs.2) The paper is very heavy in acronyms. I would recommend cutting back on some of them. For example, VO and BO could just be spelled out. SHG, on the other hand, is introduced more than once. It was established on page 4 and then again on page 8.3) The study design says that the survey instrument was based on Provan and others (2005). Please clarify what modifications that were made to this instrument to make it culturally sensitive.4) In the purposeful sample, what were the criteria for inclusion? Please say more about how the 596 were selected and recruited.5) The baseline data *were*6) Please elaborate on the meaning of this statement: "There was a reduction of scheduled caste and scheduled tribe and an increase in backward caste representation in the endline." The footnote did not really help with interpretation. Also, please elaborate on how this finding is drawn from the data.7) Going back to point 1, without a better context of how exactly the groups function, the reader will have limited understanding of the change reported in the findings. We need a mechanism of change to better assess this change.8) On page 23, the author states that, "Our study demonstrated an increase in connections not only between individuals but also an
--

	expansion of relationships between groups. Poor and traditionally lower-caste women formed SHGs and developed skills to cross boundaries and develop relationships with health providers." More context is needed (here and in the beginning). What does this change mean for the women? Why was this change needed? What were the barriers preventing this change? How did the SHGs bring about this change? There are many similar statements in the conclusion, as well, that need to be better supported and dissected.
--	--

REVIEWER	Janet Long Macquarie University, Australia
REVIEW RETURNED	17-Apr-2019

GENERAL COMMENTS	This study uses social network data collected at two time points to map changes in the social structure of key people in villages in India focusing on members of Self Help Groups and local health services. The paper is well written the Tables and graphics are of a good standard. The introduction makes a firm case for the use of social network methodology to assess the growth of social capital, a key outcome of the project. Other strengths of the paper are the large study population and the impressive response rates. I would ask the authors to respond to these points:  1. I felt there were too many acronyms – SHG and ASHA are fine, as are the ones used in the sociograms and defined by the legend, but most of the others in the manuscript (like BO and VO and GP) could be written out to make it easier to follow. 2. Methods for the social network component were incomplete.  a. Need a line about Ethical approval and how you recruited your participants. b. It was not clearly stated what the level of analysis was for the SNA. Most of the literature you cited, including the source of the validated survey you said you used (Provan, et al 2005), were organisational level studies. That is, a representative from each of the chosen agencies/organisations taking part answered on behalf of their agency. This contrasts with an individual level study in which individuals nominate their own personal ties. It seems more like a bipartite network. This needs to be made really clear. c. The nature of the tie was not given. Did you ask about referral networks, personal contacts, sources of information? It is important to know exactly what you asked. d. What was the structure of the survey? Was it a name generator (e.g., "please list five agencies / people you liaise with..."), or name interpreter (e.g., "Here's a list of agencies / people. Please select the ones with whom you work")? e. I understand that your respondents answered the survey as the holder of a particular office. ASHAs were shown to be central actors. Would be interesting to discuss this in terms of risk. If this key person leaves, will the network suffer? Is succession planning built into the role? f. Section on reciprocity requirement is good but I wondered how much data that meant you didn't include. Consider reporting that. 3. Quotes from the interviews are included but no details are given on the questions asked, methods of data capture and how you analysed it. I don't think they add much to the paper currently. They have the potential to add some context and richness to the quantitative tie data but may require a separate paper to do it.
---

	4. Page 12-13 Comparison of baseline and endpoint respondents. I see from the footnote in the Table you did a chi squared test but this is not stated clearly in the text. Just needs to be stated. I wish you well in your future research efforts JL
--	--

VERSION 1 – AUTHOR RESPONSE

Reviewer(s)' Comments to Author:

Reviewer: 1

Reviewer Name: Candace Forbes Bright

Institution and Country: East Tennessee State University, USA

Please state any competing interests or state 'None declared': None declared

Please leave your comments for the authors below

This is a fascinating study and paper and I have enjoyed the opportunity to review it. It is well-developed, but I think there are ways to improve it:

1) A broader introduction is needed. Before the getting into the role of SHGs, it would be helpful to understand their context. More information on what they are needed, how exactly they function, and any comparable initiatives would be helpful from the beginning to better understand and assess the rest of the study. Most importantly, I don't feel like I ever quite understood who was participating in the SHG and how they functioned. The methods did not clarify this either. The information in Box 1 is helpful for understanding the relationship between SHGs and VOs, but a similar introduction is needed for SHGs.

Response: We appreciate your comment and the fact that more of a context needs to be presented. We added contextual information in the Background and Study objective section (1st paragraph on page 4). We also added more of an explanation to the SHG formation in the first few sentences inside Box 1 (page 8).

2) The paper is very heavy in acronyms. I would recommend cutting back on some of them. For example, VO and BO could just be spelled out. SHG, on the other hand, is introduced more than once. It was established on page 4 and then again on page 8.

Response: We have spelled out VO, and BO.

3) The study design says that the survey instrument was based on Provan and others (2005). Please clarify what modifications that were made to this instrument to make it culturally sensitive.

Response: The structure of the instrument was modeled after Provan (2005) to develop a survey table to collect the data that would be used for analysis. The questions were developed based the UP

CMP project linkages intervention to specifically ask about how different groups coordinated health and emergency referral services. These questions were translated/reverse translated into Hindi (with attention to the local dialect), pre-tested in districts neighboring the study blocks to Uttar Pradesh to further be sensitive to local language and culture. We added a condensed version of this in the paper.

4) In the purposeful sample, what were the criteria for inclusion? Please say more about how the 596 were selected and recruited.

Response: We have further explained the recruitment and selection process (bottom paragraph in Study sample section, starting on p. 9)

5) The baseline data *were*

Response: Done!

6) Please elaborate on the meaning of this statement: "There was a reduction of scheduled caste and scheduled tribe and an increase in backward caste representation in the endline." The footnote did not really help with interpretation. Also, please elaborate on how this finding is drawn from the data.

Response: The survey had a question about the respondent's caste category, which we used to construct the caste categories in Table 2. We have added a reference to that effect in the footnote on page 15.

7) Going back to point 1, without a better context of how exactly the groups function, the reader will have limited understanding of the change reported in the findings. We need a mechanism of change to better assess this change.

Response: This comment addressed in Point 1.

8) On page 23, the author states that, "Our study demonstrated an increase in connections not only between individuals but also an expansion of relationships between groups. Poor and traditionally lower-caste women formed SHGs and developed skills to cross boundaries and develop relationships with health providers." More context is needed (here and in the beginning). What does this change mean for the women? Why was this change needed? What were the barriers preventing this change? How did the SHGs bring about this change? There are many similar statements in the conclusion, as well, that need to be better supported and dissected.

Response: Very important points! These are very complex societal issues and we have summarized some of the main points in the paper in the introduction (as explained above) and in the discussion section (2nd paragraph, p. 24).

Reviewer: 2

Reviewer Name: Janet Long

Institution and Country: Macquarie University, Australia

Please state any competing interests or state 'None declared': None declared

Please leave your comments for the authors below

This study uses social network data collected at two time points to map changes in the social structure of key people in villages in India focusing on members of Self Help Groups and local health services. The paper is well written the Tables and graphics are of a good standard. The introduction makes a firm case for the use of social network methodology to assess the growth of social capital, a key outcome of the project. Other strengths of the paper are the large study population and the impressive response rates.

I would ask the authors to respond to these points:

1. I felt there were too many acronyms – SHG and ASHA are fine, as are the ones used in the sociograms and defined by the legend, but most of the others in the manuscript (like BO and VO and GP) could be written out to make it easier to follow.

Response: Done!

2. Methods for the social network component were incomplete.

a. Need a line about Ethical approval and how you recruited your participants.

Response: We have further explained the recruitment and selection process (2nd paragraph in Study sample section, p. 10). We have added the Boston University Medical Campus IRB approval No. H-32603 at the end of the paper in Footnotes, as required by BMJ Open and also at the end of the methods section as requested by the editor..

b. It was not clearly stated what the level of analysis was for the SNA. Most of the literature you cited, including the source of the validated survey you said you used (Provan, et al 2005), were organisational level studies. That is, a representative from each of the chosen agencies/organisations taking part answered on behalf of their agency. This contrasts with an individual level study in which individuals nominate their own personal ties. It seems more like a bipartite network. This needs to be made really clear.

Response: We used the structure of the Provan survey in constructing the main table of relationships (which can be used for both SNA and ONA) and questions about whether they worked together to coordinate health services and emergency referrals. We have explained the recruitment and selection process (as listed above in (2nd paragraph in Study sample section, p. 10). All respondents in the study have "roles" to play in the villages and blocks as defined in Table 1. Therefore, potential respondents in this project had to be a member of an SHG or have a specific role or designation in the selected village and block. That is, though these are individuals, their ties do not represent themselves as individuals, but rather as SHGs or health workers or professionals in the system (thus, like one-person organizations). The level of analysis is the system as we also further categorize them into SHG structure, Government health services, RGMVP and Other (key influencers), whereas the level of observation is individuals responding on behalf of their SHG, health unit or role they embody.

c. The nature of the tie was not given. Did you ask about referral networks, personal contacts, sources of information? It is important to know exactly what you asked.

Response: We have used the Table 1 of fixed respondents and asked about whether the interviewee had a coordinated services and emergency referrals with every other type of respondent in the survey instrument. The question was: Have you coordinated health services, including emergency referrals? (See page 9) We have added this into the paper and have added “emergency” to the referral networks to be more specific.

d. What was the structure of the survey? Was it a name generator (e.g., “please list five agencies / people you liaise with...”), or name interpreter (e.g., “Here’s a list of agencies / people. Please select the ones with whom you work”)?

Response: We had a pre-fixed list of categories and/or roles with whom the respondent may have had a specific type of relationship. For example, if it was a SHG member we asked: Have you coordinated health services with the ASHA, ANM, Pradhan or village leader, primary health center medical doctor, etc...all the way down the list (with a YES/NO response). See pages 9 and 10, where this process is described.

e. I understand that your respondents answered the survey as the holder of a particular office. ASHAs were shown to be central actors. Would be interesting to discuss this in terms of risk. If this key person leaves, will the network suffer? Is succession planning built into the role?

Response: The ASHA is selected from within the local community where she lives, and therefore she tends to be in that role for a long time. As jobs are scarce in village India, there would be many women wanting that position, so finding a replacement (if needed) would not be an issue. However, it would take some time for her to go through the government designated training program (divided in about 10 modules) to be able to function effectively. So this process could be expected to impact the network to some degree. However, this impact is potentially mitigated as the SHG itself acts as a conduit of information by virtue of its group structure allowing for information sharing among members that are increasingly becoming aware of health services and entitlements. Due to limitations in the length of the paper and because this is not a current issue within the health system, we decided to not include this discussion in the paper.

f. Section on reciprocity requirement is good but I wondered how much data that meant you didn’t include. Consider reporting that.

Response: We have completed this analysis and 31% of ties are left. This is not surprising as there are very strict boundaries between different groups in Northern India in particular, and lack of reciprocity is part of the caste and poverty dimension of the SHGs. There many imbalanced relationships where lower-castes people will know higher castes but the relationship may not be reciprocated. We’ve included this in the paper with a citation of the importance of using confirmed ties as they improve the probability of establishing stronger working relationships and that was the major point of the intervention, establishing stronger coordination networks.

3. Quotes from the interviews are included but no details are given on the questions asked, methods of data capture and how you analysed it. I don’t think they add much to the paper currently.

They have the potential to add some context and richness to the quantitative tie data but may require a separate paper to do it.

Response: Thank you for the suggestion as we were struggling with whether to include them or not. We agree that adding the quotes was too much and we do plan to write a qualitative paper to be able to delve into those detailed perceptions. In view of your suggestion and our plans for a second paper, we deleted the quotes and reference to the qualitative study (as it does not affect this paper).

4. Page 12-13 Comparison of baseline and endpoint respondents. I see from the footnote in the Table you did a chi squared test but this is not stated clearly in the text. Just needs to be stated.

Response: Done

I wish you well in your future research efforts

Response: Thank you for your detailed read of the paper and the comments that helped us to further strengthen the paper!

JL

VERSION 2 – REVIEW

REVIEWER	Candace Forbes Bright East Tennessee State University, United States
REVIEW RETURNED	22-May-2019

GENERAL COMMENTS	I have re-read the article and I feel that the authors have addressed all of the critiques from the previous round of feedback. In particular, the methods are clarified. I think a little more editing is needed (for example, "data" is plural, so it should be "data are/were" not "data is/was"- I think this was only an issue once; there are some other areas that are vague, such as what "then and now" in the introduction; etc). Overall, I think this article is strong and ready for publication.
--

REVIEWER	Janet Long Macquarie University, Australia
REVIEW RETURNED	17-May-2019

GENERAL COMMENTS	Thank you for addressing my comments so well. I think the mss now reads much better. Your response about the reciprocal relationships and cultural issues with people from different castes was very interesting - a tricky methodological problem which, now I understand your rationale makes a lot of sense. Happy to recommend accept. P.S. Spotted a typo in the abstract under participants: "respondents"
--

VERSION 2 – AUTHOR RESPONSE

Reviewer(s)' Comments to Author:

Reviewer: 2

Reviewer Name: Janet Long

Institution and Country: Macquarie University, Australia

Please state any competing interests or state 'None declared': None declared

Please leave your comments for the authors below

Thank you for addressing my comments so well. I think the mss now reads much better. Your response about the reciprocal relationships and cultural issues with people from different castes was very interesting - a tricky methodological problem which, now I understand your rationale makes a lot of sense.

Happy to recommend accept.

P.S. Spotted a typo in the abstract under participants: "respondents"

Thank you...done (page 2)

Reviewer: 1

Reviewer Name: Candace Forbes Bright

Institution and Country: East Tennessee State University, United States

Please state any competing interests or state 'None declared': None declared.

Please leave your comments for the authors below

I have re-read the article and I feel that the authors have addressed all of the critiques from the previous round of feedback. In particular, the methods are clarified. I think a little more editing is needed (for example, "data" is plural, so it should be "data are/were" not "data is/was"- I think this was only an issue once; there are some other areas that are vague, such as what "then and now" in the introduction; etc). Overall, I think this article is strong and ready for publication.

Authors: 1) Removed "Then and now" (page 4); 2) Changed to data "were" (Page 9)